# Combined Naltrexone–Bupropion Therapy for Concurrent Cocaine Use Disorder and Obesity: A Case Report

**DOI:** 10.3390/reports8030174

**Published:** 2025-09-08

**Authors:** Vincenzo Maria Romeo

**Affiliations:** 1Department of Culture and Society, University of Palermo, Viale Delle Scienze, Ed. 15, 90128 Palermo, Italy; vincenzomaria.romeo@unipa.it; Tel.: +39-3405803854; 2School of Psychoanalytic and Groupanalytic Psychotherapy S.P.P.G., Via Fontana n° 1, 89131 Reggio Calabria, Italy

**Keywords:** bupropion, case report, cocaine use disorder, combined pharmacotherapy, naltrexone, obesity, weight loss

## Abstract

**Background and Clinical Significance:** Cocaine use disorder (CUD) is characterized by recurrent, cue-triggered and intrusive urges to use cocaine (craving), compulsive drug-seeking despite adverse consequences, and impaired control over intake, often co-occurring with excess weight and hedonic overeating. A dual-target rationale supports the fixed-dose naltrexone–bupropion (NB) combination: μ-opioid receptor (MOR) antagonism may mitigate opioid-facilitated mesolimbic reinforcement, while bupropion’s catecholaminergic effects and POMC activation support satiety and weight loss. **Case Presentation:** We describe a case study from an Italian outpatient setting of a 35-year-old man with a 10-year history of CUD, multiple failed detoxifications, and class I obesity (body mass index [BMI] 31 kg/m^2^) who initiated fixed-dose NB and was followed for 12 weeks under routine care. NB was associated with progressive attenuation of cue-reactive cocaine craving and improved appetite control, alongside clinically meaningful weight reduction, without psychiatric destabilization or emergent safety concerns; medication adherence remained stable. The patient maintained abstinence throughout follow-up and reported improved psychosocial functioning. Quantitatively, CCQ-B scores decreased from 7.2 at baseline to 2.1 at Week 12 (≈70% reduction), while BMI decreased from 31.0 to 25.5 kg/m^2^ (≈−17.7%), with clinically meaningful weight loss and stable adherence. **Conclusions:** This case study supports the mechanistic rationale that dual NB therapy can simultaneously attenuate cocaine craving and facilitate weight control, addressing two clinically relevant targets in CUD. Although evidence for NB in CUD remains limited and mixed across stimulant populations, this observation highlights a plausible, testable therapeutic hypothesis that integrates mesolimbic and hypothalamic pathways and may inform the design of controlled trials in patients with co-occurring CUD and obesity.

## 1. Introduction and Clinical Significance

Cocaine use disorder (CUD) is marked by recurrent, cue-triggered craving, compulsive drug-seeking despite adverse consequences, and impaired control over intake [1]. Clinically, CUD often co–occurs with excess weight and hedonic overeating, a convergence that is not merely epidemiological but reflects partially overlapping neurobiological substrates [2]. Mesolimbic reinforcement pathways—centred on the ventral tegmental area (VTA), nucleus accumbens (NAcc), and prefrontal cortex (PFC)—mediate salience attribution and reward learning [3], while hypothalamic circuits governing energy balance integrate peripheral metabolic signals with hedonic drive [4]. Within the arcuate nucleus (ARC), pro-opiomelanocortin (POMC)/cocaine-and-amphetamine-regulated transcript (CART) neurons promote satiety, whereas agouti-related peptide (AgRP)/neuropeptide-Y (NPY) neurons promote feeding; these homeostatic systems interface with mesolimbic reward and executive control, providing a plausible framework for comorbid drug reinforcement and dysregulated eating [4].

Fixed-dose NB offers a dual-target mechanistic rationale for this comorbidity. Bupropion, a norepinephrine–dopamine reuptake inhibitor (NDRI), increases tonic catecholaminergic tone and activates POMC neurons, supporting satiety and weight loss [5]. Naltrexone, a μ-opioid receptor (MOR) antagonist, removes β-endorphin-mediated autoinhibition on POMC and may attenuate opioid-facilitated mesolimbic dopaminergic bursts, thereby dampening cue-reactive reinforcement [5]. As schematized in Figure 1, naltrexone antagonises the μ-opioid receptor (MOR) and bupropion enhances catecholaminergic tone and activates POMC; hereafter, we refer to MOR throughout the text. Evidence for NB in weight management is established [5]; by contrast, signals across stimulant-use disorders are mixed—supportive in methamphetamine users (e.g., randomized data) [6] yet inconclusive in human laboratory paradigms of cocaine use [7]—highlighting a specific knowledge gap for CUD [7].

To our knowledge, fixed-dose NB has not been systematically evaluated for CUD within Italian clinical settings. Building on the above rationale, we propose that concurrent modulation of mesolimbic reinforcement and hypothalamic POMC signalling could translate into parallel clinical benefits for cocaine craving and weight. To orient the reader to this dual-target rationale, Figure 1 schematizes (A) cocaine’s mesolimbic mechanisms, (B) hypothalamic energy-balance control, and (C) the proposed combined NB actions across reward and POMC pathways. Here we present an Italian outpatient case study followed over 12 weeks of NB therapy, aiming to generate a testable clinical-mechanistic hypothesis for patients with co-occurring CUD and obesity.

## 2. Case Presentation

A 35-year-old man fulfilled DSM-5 criteria for severe cocaine use disorder (CUD) and was referred to our tertiary addiction service in March 2025 after eight unsuccessful detoxifications during the preceding decade. He reported daily intranasal use of ≈1.5 g cocaine hydrochloride, escalating expenditure, and progressive social impairment, although full-time employment was maintained. There was no history of alcohol- or opioid-use disorders, psychotic or bipolar illness, major medical diseases, or familial substance misuse. Physical examination was unremarkable except for central adiposity: weight = 95 kg, height = 1.75 m, body-mass index (BMI) = 31.0 kg/m^2^ (class I obesity). Routine laboratory work-up (blood count, biochemistry, thyroid function, fasting glucose and lipids) and ECG were normal. Baseline psychometrics showed a Hamilton Anxiety Rating Scale (HAM-A) score of 25 [8], a Hamilton Depression Rating Scale (HAM-D) score of 28 [9], and a Cocaine Craving Questionnaire-Brief (CCQ-B) score of 7.2 [10]. Urine immunoassay was positive for benzoylecgonine; all other drug screens were negative.

The patient had previously completed cognitive-behavioural therapy, contingency management, and pharmacological trials with disulfiram, modafinil, sertraline, and quetiapine without sustained abstinence. Because of persistent cue-induced craving, weight gain during prior recovery attempts, and high relapse risk, we proposed off-label treatment with the fixed-dose NB combination, which targets μ-opioid and mesolimbic dopaminergic pathways implicated in both cocaine craving and energy-balance regulation. After detailed discussion of benefits, risks, alternatives, and the experimental nature of NB, written informed consent was obtained in accordance with CARE standards (Appendix A) and institutional ethics policy (exempt review) [11].


**Therapeutic intervention and timeline**


NB was started at 8/90 mg day^−1^ (week 0) and titrated every two weeks—16/180 mg day^−1^ (week 2) and 24/270 mg day^−1^ (week 4)—to the obesity-licensed target of 32/360 mg day^−1^ by week 8. Figure 2 depicts the 12-week clinical timeline, including NB titration, urine toxicology results, scheduled psychotherapy visits, and adverse events.

Weekly 50-minute cognitive-behavioural sessions focused on relapse prevention were continued, and dietetic counselling aimed at a 500 kcal daily deficit plus ≥ 150 min moderate exercise per week was introduced. Craving (CCQ-B) and mood (HAM-A, HAM-D) were assessed bi-weekly; adverse effects were monitored with the UKU Side-Effect Rating Scale, patient version (UKU-SERS-Pat) [12]. Medication adherence was reinforced by brief motivational interviewing, pill counts, and twice-weekly electronic reminders. 


**Outcomes**


Throughout the 12-week observation, the patient maintained uninterrupted biochemical abstinence. CCQ-B scores dropped by 70%, accompanied by clinically meaningful reductions in anxiety (–52%) and depression (–46%).The week-by-week therapeutic timeline and clinical measures (NB dose, urine cocaine, CCQ-B, HAM-A, HAM-D, and BMI) are summarised in Table 1. An expanded visualisation of the craving trajectory is provided in Appendix A. BMI fell progressively to 25.5 kg/m^2^ (–17.7% body-weight change), facilitated by improved diet and increased physical activity logged daily. Transient nausea, insomnia, and dry mouth emerged during up–titration but resolved within ten days; the UKU-SERS-Pat total decreased from 20 to 5 [12]. A detailed safety summary is provided in Table 2 (adverse-event profile and UKU-SERS-Pat totals across time points).

**Table 1 reports-08-00174-t001:** Therapeutic Timeline and Clinical Measures.

Week	NB Dose (mg)	Urine Cocaine	CCQ-B	HAM-A	HAM-D	BMI (kg/m^2^)
0	8/90	Positive	7.2	25	28	31.0
4	16/180	Negative	4.1	18	22	28.7
8	32/360	Negative	2.9	14	17	26.4
12	32/360	Negative	2.1	12	15	25.5

**Legend:** Week = time from NB initiation; NB = naltrexone–bupropion daily dose; CCQ-B = Cocaine Craving Questionnaire-Brief; HAM-A = Hamilton Anxiety Rating Scale; HAM-D = Hamilton Depression Rating Scale; BMI = body-mass index.

**Table 2 reports-08-00174-t002:** Tolerability profile during 12-week NB therapy. Part A. Adverse-event (AE) profile during up-titration. Part B. UKU-SERS-Pat total score over time.

**Part A**					
**Adverse Event**	**Week of Onset**	**Duration (Days)**	**Severity**	**Action Taken**	**Outcome**
**Nausea**	Week 1	7	Mild	Dose maintained; small, frequent meals; hydration	Resolved without sequelae
**Insomnia**	Week 2	10	Mild	Morning dosing; caffeine reduction; CBT-based sleep hygiene	Resolved without sequelae
**Dry mouth**	Week 2–3	10	Mild	Hydration; sugar-free gum; oral care	Resolved without sequelae
**Part B**					
**Time point**			**UKU-SERS-Pat total**		
**Week 0 (baseline)**			20		
**Week 4**			12		
**Week 8**			8		
**Week 12**			5		

Legend: Severity graded by patient-rated UKU-SERS-Pat as Mild/Moderate/Severe; duration is the number of consecutive days with symptoms. Abbreviations: UKU-SERS-Pat, Udvalg for Kliniske Undersøgelser Side Effect Rating Scale—patient version; CBT, cognitive-behavioural therapy; NB, naltrexone–bupropion.

No laboratory or ECG abnormalities occurred, and adherence exceeded 95%. The patient reported enhanced energy, self-efficacy, and social engagement, and elected to continue NB under quarterly follow-up. A six-month interim telephone contact confirmed ongoing abstinence, stable weight, and absence of late-emerging adverse events. Figure 3 illustrates the trajectories of craving (CCQ-B), anxiety (HAM-A), depression (HAM-D), and BMI across the 12-week treatment.

## 3. Discussion

This case study suggests that the fixed-dose NB combination may concurrently attenuate cocaine craving and facilitate clinically meaningful weight loss in a patient with cocaine use disorder (CUD) and class I obesity. Current guidance underscores that no medications are approved for CUD; bupropion may be considered (low-certainty evidence), and NB is conditionally recommended only for amphetamine-type stimulant use disorder based on emerging data—highlighting the off-label status of NB for CUD and the need for controlled studies in this population [13].

Our observations align partly with pharmacologic signals from stimulant-use populations. In the two-stage, multisite NEJM trial (ADAPT-2), extended-release naltrexone plus bupropion improved response rates in methamphetamine use disorder versus placebo, supporting the conceptual rationale for dual μ-opioid antagonism and catecholaminergic modulation to reduce stimulant reinforcement [14]. Conversely, a human laboratory study in non-treatment-seeking adults found that NB did not acutely reduce cocaine self-administration, suggesting that patient motivation, clinical context, outcome windows (acute self-administration vs weeks-long abstinence), and concurrent psychosocial care may moderate real-world effectiveness [15]. On the satiety axis, a disinhibition-plus-activation motif (naltrexone on MOR; bupropion on POMC) could be functionally superadditive for weight control, whereas anti-craving effects are more plausibly additive across distinct targets.

Beyond cocaine outcomes, psychiatric tolerability and weight effects with NB are relevant to our patient. In a double-blind trial for binge-eating disorder, NB produced clinically meaningful weight loss with acceptable psychiatric safety—converging with our patient’s BMI reduction to 25.5 kg/m^2^ (overweight range) and improvement in anxiety/depression ratings during 12 weeks of treatment [16]. From a systems perspective, these findings fit a broader evidence base in which pharmacotherapy is often most useful when integrated with behavioural strategies. Comparative and meta-analytic work in CUD shows the strongest and most consistent benefits for contingency management, while pharmacotherapies yield mixed effects across endpoints and subgroups—reinforcing the value of the combined approach we used (NB plus weekly CBT and lifestyle counselling) [17].

Of note, our results converge with a recent case report using the GLP-1 receptor agonist semaglutide in a patient with obesity and cocaine abuse, which documented parallel reductions in weight and cocaine craving over 12 weeks. Although mechanistically distinct, both approaches target metabolic-reward cross-talk, lending plausibility to dual-target strategies for patients with overlapping obesity and stimulant-use phenotypes [18].

Limitations: This single-patient, open-label observation without a comparator cannot infer causality or pharmacodynamic interaction (additivity/synergy). Outcomes relied on patient-reported craving measures and short (12-week) follow-up; behavioural confounders (dietary changes, activity) cannot be fully excluded. Biomarkers of opioid or catecholaminergic tone were not collected. These limitations warrant cautious interpretation and motivate controlled trials [11,13,19].

Clinical Significance: This case study describes, for the first time in Italy, the use of the fixed-dose NB combination to treat a patient with severe cocaine use disorder (CUD) complicated by class I obesity. Over 12 weeks, the regimen produced (i) a ≈ 70% reduction in cocaine craving with continuous biochemical abstinence, (ii) clinically meaningful improvements in anxiety and depression, and (iii) a substantial BMI reduction from 31.0 to 25.5 kg/m^2^ (overweight range), all with only transient, mild adverse effects. The case highlights the dual therapeutic potential of this pharmacological pairing—simultaneous modulation of μ-opioid and mesolimbic dopaminergic pathways—to address two high-risk, often co-existing targets (craving-driven relapse and weight gain) that current CUD treatments fail to cover. These findings support further controlled trials of NB as an innovative, mechanism-based option within multidisciplinary care for relapsing, overweight individuals with CUD.

Post-treatment safety considerations after rapid weight loss: Given the ≈17.7% weight reduction over 12 weeks, post-treatment safety warrants explicit consideration. Rapid loss increases the risk of biliary complications (e.g., gallstones) [20], and may exacerbate lean-mass depletion if protein intake and resistance exercise are insufficient [21]. Adaptive endocrine responses (e.g., reduced leptin, increased ghrelin) can favour weight regain after discontinuation, necessitating maintenance strategies (dietary quality, progressive physical activity, behavioural supports) [22]. Routine monitoring should include liver function tests (naltrexone), blood pressure and neuropsychiatric status, and counselling on seizure-threshold-lowering factors relevant to bupropion. Patients should be educated to seek evaluation for biliary symptoms (post-prandial right-upper-quadrant pain, nausea). A planned maintenance phase (continued NB per label for weight, or taper with behavioural consolidation) may mitigate rebound in appetite and relapse risk.

Clinical implications and research needs: NB may be considered as an off-label, mechanism-based option for overweight individuals with CUD within multidisciplinary programs, with careful safety monitoring (e.g., seizure risk with bupropion) and integration with evidence-based behavioural treatments. Priority research should include randomized, adequately powered trials in CUD that (i) compare NB with guideline-suggested options and with contingency management augmentation, (ii) incorporate objective adherence monitoring and frequent toxicology, (iii) stratify by obesity/metabolic status and baseline use severity, and (iv) assess durability beyond 6–12 months. Head-to-head or adaptive designs that include GLP-1 receptor agonists could also elucidate whether metabolic-reward targeting improves outcomes in metabolically vulnerable subgroups.

### Mechanistic Synergy (Hypothesis) Across Satiety and Reinforcement Pathways

As schematized in Figure 1, naltrexone antagonises the μ-opioid receptor (MOR) while bupropion enhances catecholaminergic tone and activates POMC. On the satiety axis, this disinhibition-plus-activation motif (naltrexone on MOR; bupropion on POMC) could be functionally superadditive for weight control, because POMC signalling is simultaneously freed from β-endorphin autoinhibition and driven by catecholaminergic activation. By contrast, anti-craving effects are more plausibly additive across partly distinct targets (opioid modulation and catecholaminergic tone) in cocaine reinforcement. Importantly, formal synergy metrics cannot be inferred from a single-patient observation and would require multi-arm designs to disentangle additivity from supra-additivity. We therefore frame synergy as a hypothesis-generating construct restricted to the satiety pathway, while adopting an additive stance for anti-craving mechanisms pending controlled testing.

## 4. Conclusions

In this single-patient case study, fixed-dose NB was associated with progressive reductions in cue-reactive cocaine craving, sustained biochemical abstinence, and clinically meaningful weight loss over 12 weeks, with only transient, mild adverse effects. As schematized in Figure 1, NB appears to concurrently target mesolimbic reinforcement and hypothalamic POMC signalling, offering a plausible mechanistic rationale for the dual clinical trajectory observed. These observations are hypothesis-generating and require controlled testing. Given the off-label use, single-case design, short on-treatment observation, and potential confounding by concurrent cognitive-behavioural therapy and lifestyle counselling, causal inference and generalizability are limited. Cautious clinical use is warranted, with attention to known risks (e.g., lowered seizure threshold with bupropion; opioid blockade with naltrexone, including perioperative analgesia planning and the risk of precipitated withdrawal with undisclosed opioid use). Prospective, randomized studies with longer follow-up, standardized craving and functional outcomes, objective adherence and toxicology monitoring, and stratification by metabolic phenotype are needed to determine efficacy, safety, and durability, and to clarify whether treatment effects are additive or functionally synergistic across satiety and reinforcement pathways.

## Figures and Tables

**Figure 1 reports-08-00174-f001:**
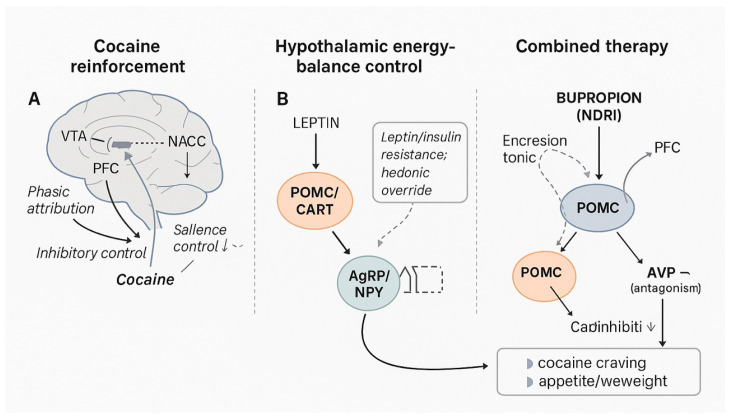
Integrated neurobiological model of cocaine reinforcement, hypothalamic energy-balance control, and dual NB action. Legend: Panel (A) (Cocaine reinforcement): Cocaine blocks the dopamine transporter (DAT) on mesolimbic neurons, producing phasic dopamine surges in the ventral tegmental area (VTA)–nucleus accumbens (NAcc) pathway, with impaired top-down inhibitory control from the prefrontal cortex (PFC) and stress/glutamatergic plasticity that increase salience attribution to drug cues. Panel (B)(Energy-balance control): Hypothalamic arcuate nucleus (ARC) integrates peripheral metabolic signals. The POMC/CART anorexigenic pathway opposes the orexigenic AgRP/NPY neurons; leptin/insulin signalling modulates these populations and interfaces with hedonic reward (NAcc), allowing “hedonic override” under resistance states (Combined therapy): Bupropion (NDRI) raises tonic catecholaminergic tone and activates POMC; naltrexone (μ-opioid receptor antagonist) removes β-endorphin-mediated autoinhibition on POMC and dampens opioid-facilitated mesolimbic dopamine. The combined effect plausibly reduces cocaine cue-reactivity/craving and supports appetite/weight reduction via sustained POMC signalling. Abbreviations: DAT, dopamine transporter; VTA, ventral tegmental area; NAcc, nucleus accumbens; PFC, prefrontal cortex; ARC, arcuate nucleus; POMC, pro-opiomelanocortin; CART, cocaine-and-amphetamine-regulated transcript; AgRP, agouti-related peptide; NPY, neuropeptide Y; NDRI, norepinephrine–dopamine reuptake inhibitor; MOR, μ-opioid receptor.

**Figure 2 reports-08-00174-f002:**
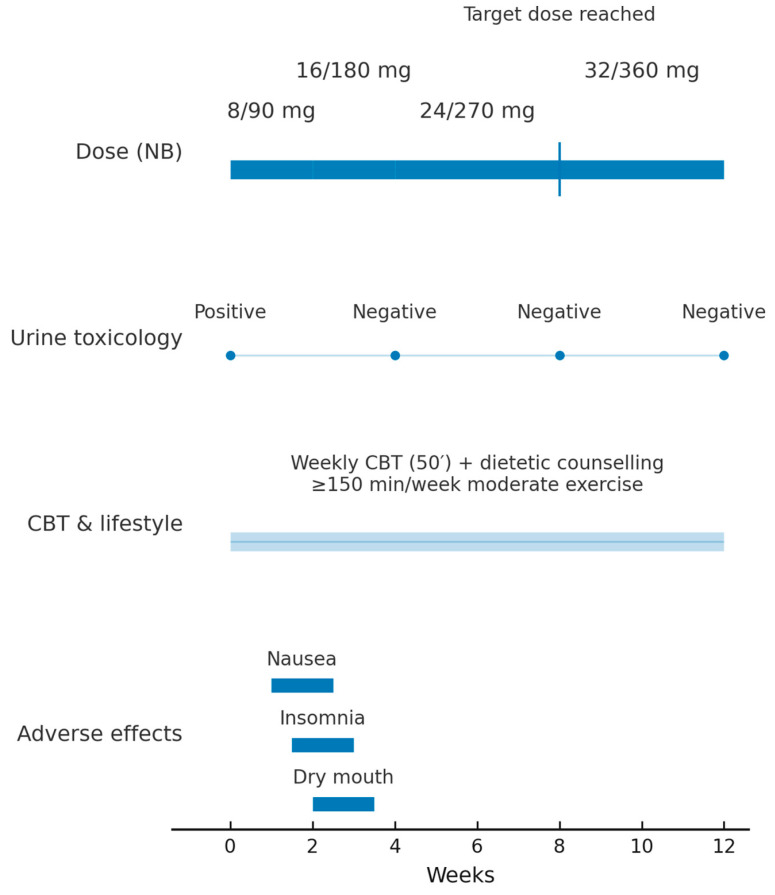
Twelve-week clinical timeline of fixed-dose NB therapy Legend. Timeline from baseline (Week 0) to Week 12 showing NB dose escalation (8/90 → 16/180 → 24/270 → 32/360 mg; target dose reached at Week 8), urine benzoylecgonine results (Positive at Week 0; Negative at Weeks 4, 8, and 12), continuous weekly CBT and lifestyle counselling (≥150 min/week moderate exercise), and transient adverse effects during up-titration (nausea, insomnia, dry mouth). Abbreviations: NB, naltrexone–bupropion; CBT, cognitive-behavioural therapy.

**Figure 3 reports-08-00174-f003:**
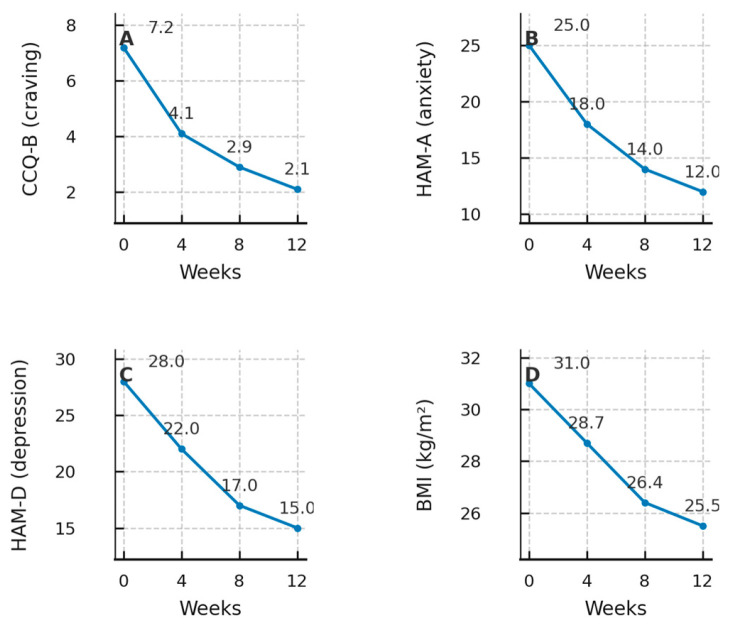
Outcome trajectories over 12 weeks of NB therapy. Legend. Four-panel composite showing longitudinal changes at Weeks 0, 4, 8, and 12. Panel (**A**): CCQ-B (craving); Panel (**B**): HAM-A (anxiety); Panel (**C**): HAM-D (depression); Panel (**D**): Body mass index (BMI). Points mark observed values and lines connect successive timepoints; numeric values are displayed near markers for readability. Abbreviations: CCQ-B, Cocaine Craving Questionnaire-Brief; HAM-A, Hamilton Anxiety Rating Scale; HAM-D, Hamilton Depression Rating Scale; BMI, body-mass index.

## Data Availability

All data supporting the findings of this study are contained within the article. De-identified source data (e.g., scale scores and time-stamped toxicology results) can be provided by the corresponding author upon reasonable request, subject to privacy and institutional safeguards. No publicly archived datasets were generated.

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
