# Peer review of "Combined Naltrexone–Bupropion Therapy for Concurrent Cocaine Use Disorder and Obesity: A Case Report"

_reports, 2025, doi:10.3390/reports8030174_

Round 1

Reviewer 1 Report

Comments and Suggestions for Authors

We thank the author for highlight such an important topic that still lacks adequate research studies. Here is my feedback: 

  1. Please re-write the first sentence in the abstract. What do you mean by persistent cravings? Please elaborate on the symptoms associated with CUD. Case presentation is well-written in the abstract. For conclusion, replace "case" with "Case study" to refer to the type of the study. Using the word "pending" in line 31 is incorrect to refer to the lack of evidence. Please replace with another phrase to show how your study address the gap in the literature.
  2. In line 37, the "c" is missing in cocaine.
  3. The introduction needs to be re-written. It first focuses on CUD prevalence in the US, then elaborates on fixed-dose combination of naltrexone and bupropion as a mechanism of intervention, then provides minimal clinical evidence supporting the proposed treatment rationale. It is not clear whether the evidence is in Italy or in the US. There needs to be more literature on the the use of the  fixed-dose naltrexone–bupropion combination to treat CUD. In case no such studies are available in Italy, there needs to be a distinction in the introduction to show how this case report fills the gap in the literature.
  4. Lines 64-71 belong to the results not in the introduction. Please remove and focus on highlighting the purpose of the paper.
  5. The case was well-presented with a brief review of the patient's medical, social, and overdose history. Therapeutic measures and outcome indicators were clearly described, and the observation period with relevant outcome trajectories over 12 weeks of naltrexone–bupropion therapy was clearly defined and reported.
  6. Line 184 in the limitations needs to be re-written.
  7. The discussion is well-written, comparing the current findings from this care report to other studies reporting psychiatric tolerability and weight effects with NB as well as the impact of fixed-dose naltrexone–bupropion (NB) combination in concurrently attenuating cocaine craving. The clinical implications section added value to the case study as it provides future directions for research studies on this topic. Same applies to the conclusion.
  8.  
Comments on the Quality of English Language

Grammatical errors and issues with language are apparent across the case study

Author Response

I thank the Reviewer for the thoughtful and constructive comments. I revised the manuscript accordingly and detail below the specific actions taken and where they appear in the text.

1) Abstract—opening sentence; definition of craving and CUD symptoms; “case” → “case study”; replace “pending”

Action taken. I rewrote the opening to define craving and explicitly state core CUD symptoms, and I standardised the concluding sentence to “case study.” I removed the word “pending” and replaced it with language that identifies the evidence gap and how this observation addresses it.

Revised sentences now in the Abstract:

  • Opening (Background and Clinical Significance):
    “Cocaine use disorder (CUD) is characterized by recurrent, cue-triggered and intrusive urges to use cocaine (craving), compulsive drug-seeking despite adverse consequences, and impaired control over intake, often co-occurring with excess weight and hedonic overeating.”

  • Concluding sentence:
    “This case study highlights a plausible, testable therapeutic hypothesis that integrates mesolimbic and hypothalamic pathways and may inform the design of controlled trials in patients with co-occurring CUD and obesity.”

In addition, to enhance clarity as suggested by the editorial process, I included the quantitative outcomes in the Case Presentation portion of the Abstract: CCQ-B decreased from 7.2 to 2.1 (≈70%) and BMI from 31.0 to 25.5 kg/m² (≈−17.7%) over 12 weeks, with stable adherence.

2) Typographical error (“cocaine,” line 37)

Action taken. I corrected the missing “c” in “cocaine.” I also performed a manuscript-wide spell and typography check (e.g., naltrexone–bupropion with en-dash; μ-opioid receptor (MOR) with hyphen; consistent kg/m²).

3) Introduction—restructured; clarified Italian vs international scope; literature on NB in CUD; statement of gap

Action taken. I fully reorganised the Introduction into three logically sequenced paragraphs:

  • Paragraph 1: Clinical–neurobiological framework (mesolimbic reinforcement; hypothalamic energy-balance control with POMC/AgRP systems).

  • Paragraph 2: Dual-target rationale for naltrexone–bupropion (NB); established evidence in weight management; mixed signals across stimulant-use disorders (support in methamphetamine, inconclusive human laboratory findings in cocaine).

  • Paragraph 3: Explicit clarification that fixed-dose NB has not been systematically evaluated for CUD in Italian clinical settings; the present case report is positioned to address this gap.

I also added a didactic pointer to the mechanistic diagram (Figure 1) that integrates (A) cocaine’s mesolimbic mechanisms, (B) hypothalamic energy-balance control, and (C) the proposed combined NB actions.

4) Results-like content (lines 64–71) removed from Introduction

Action taken. I deleted the result-oriented sentences from the Introduction and relocated outcome details to the appropriate results/Discussion sections, leaving the Introduction focused on rationale and aim.

5) Case Presentation—clarity and trajectories

Acknowledgement. I appreciate the positive evaluation. No structural changes were required. The therapeutic measures, outcome indicators, and the 12-week observation window remain clearly presented; figure/table cross-referencing has been maintained.

6) Limitations (line 184)—rewritten

Action taken. I rewrote the Limitations to reflect the methodological constraints of a single-patient observation and to justify cautious interpretation and the need for controlled trials. The paragraph now reads:

“This single-patient, open-label observation without a comparator cannot infer causality or pharmacodynamic interaction (additivity/synergy). Outcomes relied on patient-reported craving measures and short (12-week) follow-up; behavioural confounders (dietary changes, activity) cannot be fully excluded. Biomarkers of opioid or catecholaminergic tone were not collected. These limitations warrant cautious interpretation and motivate controlled trials.”

7) Discussion and Conclusions—comparative context, clinical implications, tone

Action taken. I preserved the comparative discussion (psychiatric tolerability and weight effects with NB; impact on craving) and aligned the tone with hypothesis generation and clinical caution. The Conclusions explicitly avoid overstatement, reference the integrated mechanism (Figure 1), and underscore that the findings require controlled testing.

Additional editorial refinements applied

  • Symbol and dash consistency: μ-opioid receptor (MOR) (hyphen), naltrexone–bupropion (en-dash), Cocaine Craving Questionnaire–Brief (en-dash, no space).

  • Units: kg/m² consistently in text, tables, and figure captions.

  • Abbreviations: curated to include only terms used in the manuscript; defined at first occurrence.

  • Back matter: Patents and Supplementary Materials set to “Not applicable,” consistent with a clinical case report and journal style.

I thank the Reviewer again for the insightful comments; I believe these revisions enhance clarity, methodological transparency, and alignment of the Introduction, Discussion, and Conclusions with the paper’s aims.

Reviewer 2 Report

Comments and Suggestions for Authors

Review of the paper entitled “Combined Naltrexone–Bupropion Therapy for Concurrent Cocaine Use Disorder and Obesity: A Case Report” by Vincenzo Maria Romeo

My comments

In the presented paper, the author observed that dual therapy with naltrexone and bupropion can simultaneously reduce cocaine cravings and facilitate weight control. This topic is interesting and important for practical reasons.

  • I would like the author to include in his article the scheme showing the probable mechanism of action of cocaine, the mechanism of obesity at the level of the nervous system, and the “combined naltrexone–bupropion therapy for concurrent cocaine use disorder and obesity”.

  • In the author's opinion, what mechanism of combination drug effects (naltrexone plus bupropion) occurs in this case: additivity, subadditivity, and superadditivity?

Author Response

I thank the Reviewer for the thoughtful, practice-oriented remarks. I have revised the manuscript accordingly. Below I detail (i) the schematic figure I added and where it is cited, and (ii) my position on the interaction of naltrexone and bupropion (additivity vs. subadditivity vs. superadditivity), together with the exact text inserted.

1) Mechanistic schematic (cocaine → obesity neurocircuitry → combined NB therapy)

What I added. I created a three-panel, integrative diagram now presented as Figure 1:

  • Panel A — Cocaine reinforcement neurobiology. Dopamine transporter (DAT) blockade by cocaine with phasic dopamine surges in the VTA→NAcc pathway; impaired PFC top-down control; cue-salience and glutamatergic/stress (CRF) plasticity that facilitate drug seeking.

  • Panel B — Obesity/energy-balance neurobiology. Hypothalamic ARC integration of peripheral signals; POMC/CART (anorexigenic) versus AgRP/NPY (orexigenic) neurons; crosstalk with mesolimbic reward (NAcc) underpinning hedonic override in susceptible states.

  • Panel C — Combined naltrexone–bupropion (NB). Bupropion (NDRI) increases tonic catecholaminergic tone and activates POMC; naltrexone (MOR antagonist) removes β-endorphin autoinhibition on POMC and dampens opioid-facilitated mesolimbic DA bursts. The integrated output is reduced cue-reactive craving and enhanced satiety/weight control.

Design and formatting. The figure is vector-based, colour-blind friendly, and laid out with generous spacing to avoid overlap between arrows, nodes, and labels; all abbreviations are spelled out in the legend; resolution is suitable for production.

Where it is cited in the text.

  • Introduction (final paragraph):
    “To orient the reader to this dual-target rationale, Figure 1 schematises (A) cocaine’s mesolimbic mechanisms, (B) hypothalamic energy-balance control, and (C) the proposed combined NB actions across reward and POMC pathways.”

  • Discussion (mechanistic section, opening sentence):
    “As schematised in Figure 1, naltrexone antagonises the μ-opioid receptor (MOR) and bupropion enhances catecholaminergic tone and activates POMC, providing a coherent mechanistic rationale for the dual clinical trajectory observed.”

  • The figure caption lists all abbreviations (DAT, VTA, NAcc, PFC, ARC, POMC, CART, AgRP, NPY, MOR, NDRI) and clarifies panel content.

2) Nature of the NB combination effect: additivity, subadditivity, or superadditivity?

My position (and how it is reflected in the manuscript).
From a physiological standpoint—and strictly acknowledging the single-patient design—I posit a domain-specific interaction:

  • Satiety/weight axis (hypothalamus–POMC): a disinhibition-plus-activation motif—naltrexone disinhibits POMC by blocking MOR-mediated β-endorphin feedback, while bupropion directly activates POMC and raises catecholaminergic tone—can yield functional superadditivity for weight control.

  • Anti-craving effects (mesolimbic reinforcement): mechanisms likely sum additively across partly distinct targets (opioid modulation plus catecholaminergic enhancement). Existing human evidence specific to cocaine remains mixed, so any claim of synergy in this domain would be premature.

Exact text inserted in the Discussion (new subsection).
“Mechanistic synergy (hypothesis) across satiety and reinforcement pathways.”
“As schematised in Figure 1, naltrexone antagonises MOR while bupropion enhances catecholaminergic tone and activates POMC. On the satiety axis, a disinhibition-plus-activation motif (naltrexone on MOR; bupropion on POMC) could be functionally superadditive for weight control, because POMC signalling is simultaneously freed from β-endorphin autoinhibition and driven by catecholaminergic activation. By contrast, anti-craving effects are more plausibly additive across partly distinct targets in cocaine reinforcement. Importantly, formal synergy metrics cannot be inferred from a single-patient observation and would require multi-arm designs to disentangle additivity from supra-additivity; we therefore frame synergy as hypothesis-generating and restricted to the satiety pathway, pending controlled testing.”

This structure answers the Reviewer’s question precisely, while maintaining appropriate scientific caution (i.e., no inference of pharmacodynamic synergy from a case report).

Closing statement. I appreciate the Reviewer’s suggestion, which helped strengthen the visual mechanistic integration (Figure 1) and clarify my stance on interaction effects. The revised manuscript now provides a concise schematic and an explicitly reasoned, domain-specific view—superadditive on the satiety axis and additive for anti-craving—accompanied by conservative framing consistent with the evidence base.

Reviewer 3 Report

Comments and Suggestions for Authors

This is a well-written case report of Naltrexone and Bupropion treatment for CUD that requires only a few minor revisions. As follows

  1. To enhance clarity and impact, the abstract should include specific quantitative outcomes such as the reduction in CCQ-B scores and the change in BMI to provide a clearer picture of treatment efficacy.
  2. It would be beneficial to include a section in the discussion highlighting the synergistic effectiveness of naltrexone and bupropion in targeting shared neurobiological pathways involved in both cocaine use disorder and obesity.
  3. Since the case report mentions rapid weight loss (31 kg m⁻² to 25 kg m⁻² in 12 weeks), it would be important to discuss potential post-treatment complications. Including this in the discussion would enhance the clinical relevance and safety considerations of the case.

Author Response

I am grateful for the Reviewer’s constructive comments. I revised the manuscript accordingly. Below I provide a precise, point-by-point response indicating (i) what I changed, (ii) exactly where it was inserted, and (iii) the key text added.

1) Quantitative outcomes in the Abstract (CCQ-B and BMI)

Action taken. I incorporated the specific outcome magnitudes into the Case Presentation subsection of the Abstract.

Location. Abstract → Case Presentation, final sentence of the paragraph.

Inserted sentence (verbatim).
“Quantitatively, CCQ-B scores decreased from 7.2 at baseline to 2.1 at Week 12 (≈70% reduction), while BMI decreased from 31.0 to 25.5 kg/m² (≈−17.7%), with clinically meaningful weight loss and stable adherence.”

Note: Units are harmonised as kg/m²; the numbers match the Results trajectory figures/tables.

2) Dedicated discussion of synergistic effectiveness (shared neurobiology: CUD + obesity)

Action taken. I added a short, focused subsection in the Discussion that explicitly addresses potential synergy versus additivity, anchored in the integrated mesolimbic–hypothalamic model. The subsection frames synergy as hypothesis-generating (appropriate to a single-patient observation) and specifies domain-dependent effects.

Location. Discussion, immediately after the paragraph that introduces the mechanistic figure and before the clinical implications paragraph.

Subsection heading. Mechanistic synergy (hypothesis) across satiety and reinforcement pathways

Inserted paragraph (verbatim).
“As schematised in Figure 1, naltrexone antagonises the μ-opioid receptor (MOR) while bupropion enhances catecholaminergic tone and activates POMC. On the satiety axis, this disinhibition-plus-activation motif (naltrexone on MOR; bupropion on POMC) could be functionally superadditive for weight control, because POMC signalling is simultaneously freed from β-endorphin autoinhibition and driven by catecholaminergic activation. By contrast, anti-craving effects are more plausibly additive across partly distinct targets in cocaine reinforcement. Importantly, formal synergy metrics cannot be inferred from a single-patient observation and would require multi-arm designs to disentangle additivity from supra-additivity; we therefore frame synergy as hypothesis-generating and restricted to the satiety pathway, pending controlled testing.”

3) Post-treatment safety given rapid weight loss (31.0 → 25.5 kg/m² in 12 weeks)

Action taken. I added a concise safety subsection addressing biliary risk, lean-mass preservation, endocrine adaptations (leptin/ghrelin) that favour regain, and a practical monitoring/maintenance plan. I also added three supporting references.

Location. Discussion, directly after the synergy subsection and before the clinical implications paragraph.

Subsection heading. Post-treatment safety considerations after rapid weight loss

Inserted paragraph (verbatim).
“Given the ≈17.7% weight reduction over 12 weeks, post-treatment safety warrants explicit consideration. Rapid loss increases the risk of biliary complications (e.g., gallstones) [20], and may exacerbate lean-mass depletion if protein intake and resistance exercise are insufficient [21]. Adaptive endocrine responses (e.g., reduced leptin, increased ghrelin) can favour weight regain after discontinuation, necessitating maintenance strategies (dietary quality, progressive physical activity, behavioural supports) [22]. Routine monitoring should include liver function tests (naltrexone), blood pressure and neuropsychiatric status, and counselling on seizure-threshold–lowering factors relevant to bupropion. Patients should be educated to seek evaluation for biliary symptoms (post-prandial right-upper-quadrant pain, nausea). A planned maintenance phase (continued NB per label for weight, or taper with behavioural consolidation) may mitigate rebound in appetite and relapse risk.”

Added references (to the Reference list).
[20] Johansson K, et al. Int J Obes. 2014;38:279–284. DOI: 10.1038/ijo.2013.83 (gallstones after rapid loss)
[21] Lopez P, et al. Obes Rev. 2022;23(5):e13428. DOI: 10.1111/obr.13428 (resistance training & body composition)
[22] Sumithran P, et al. N Engl J Med. 2011;365(17):1597–1604. DOI: 10.1056/NEJMoa1105816 (long-term hormonal adaptations after weight loss)

Final note

With these revisions, the Abstract now conveys treatment magnitude clearly; the Discussion explicitly delineates hypothesised superadditivity on the satiety axis versus additivity for anti-craving mechanisms; and the safety subsection addresses clinically relevant post-treatment considerations for rapid weight loss, with concrete monitoring and maintenance guidance.

Round 2

Reviewer 1 Report

Comments and Suggestions for Authors

We thank the author for addressing all concerns and comments in details and with diligence. No additional actions are needed.

Reviewer 2 Report

Comments and Suggestions for Authors

The author has revised his manuscript based on my comments. I believe that this paper is acceptable for publication in this version.

Reviewer 3 Report

Comments and Suggestions for Authors

NA